# Oral Administration of Human-Gut-Derived *Prevotella histicola* Improves Sleep Architecture in Rats

**DOI:** 10.3390/microorganisms11051151

**Published:** 2023-04-28

**Authors:** Dae Wui Yoon, Inkyung Baik

**Affiliations:** 1Department of Biomedical Laboratory Science, Jungwon University, Goesan-gun 28204, Chungcheongbuk-do, Republic of Korea; ydw@jwu.ac.kr; 2Department of Foods and Nutrition, College of Science and Technology, Kookmin University, Seoul 02707, Republic of Korea

**Keywords:** gastrointestinal microbiome, probiotics, sleep aids, animal experimentation, *Prevotella*, electroencephalography

## Abstract

(1) Background: The human gut microbiome may regulate sleep through the gut–brain axis. However, the sleep-promoting effects of gut microbiota remain unclear. (2) Methods: We obtained sleep–wake profiles from 25 rats receiving *P. histicola* (*P. histicola* group), 5 rats receiving *P. stercorea* (*P. stercorea* group), 4 rats not receiving bacteria (No administration group), and 8 rats receiving *P. histicola* extracellular vesicles (EV) (EV group) during the baseline, administration, and withdrawal periods. (3) Results: The *P. histicola* group showed increased total sleep, rapid eye movement (REM) sleep, and non-rapid eye movement (NREM) sleep time during the administration and withdrawal periods; on the last day of administration, we found significant increases of 52 min for total sleep (*p* < 0.01), 13 min for REM sleep (*p* < 0.05), and 39 min for NREM sleep (*p* < 0.01) over the baseline. EV administration also increased NREM sleep time on Day 3 of administration (*p* = 0.05). We observed a linear trend in the dose–response relationship for total sleep and NREM sleep in the *P. histicola* group. However, neither the no-administration group nor the *P. stercorea* group showed significant findings. (4) Conclusions: Oral administration of probiotic *P. histicola* may improve sleep and could be a potential sleep aid. Further rigorous evaluations for the safety and efficacy of *P. histicola* supplementation are warranted.

## 1. Introduction

Potential roles of gut microbiota in the gut–brain axis have been suggested [1]. Particularly, gut microbiota could be linked to sleep–wake states through gamma amino butyric acid (GABA) and serotonin production, circadian rhythm modulation, and altered blood–brain-barrier permeability by commensal microbiota [1,2]. An association between inadequate sleep and imbalance in the gut microbiota, called *dysbiosis*, has been reported in animal models [3,4,5,6] and humans [4,5,6,7,8,9,10]. These studies have shown decreased microbiota richness and diversity [4,5,8,9,11] and altered bacterial composition at the phylum, family, or genus levels [3,5,6,7,8,10,11] in association with inadequate sleep or insomnia. In particular, increased abundance of the phylum *Firmicutes* and decreased abundance of the phylum *Bacteroidetes* [3,11] or an increased *Firmicutes* to *Bacteroidetes* ratio [6,7] have been observed.

However, findings at the genus level were inconsistent or even conflicting [5,8,9,10,11]. In addition, the causal relationship between the altered composition of a particular bacterium and inadequate sleep is questionable; it is unclear whether the bacterium is a causal factor or inadequate sleep alters the bacterial composition as a sleep-inducing signal to compensate for inadequate sleep. A few studies have examined the effects of probiotic supplementation with a single species, such as *Lactobacillus* or *Bifidobacterium*, on the sleep states in animals [12,13] and humans [14,15].

The genus *Prevotella* includes more than 50 species, belonging to the phylum *Bacteroidetes*, that are found in the human oral cavity, small intestine, and vagina [16]. It has been reported that *Prevotella* species, such as *P. copri*, *P. gingivalis, P. melanogenica*, and *P. nigrescens*, are associated with several inflammatory diseases, but all species may not with different properties [17]. Recent animal studies have suggested the potential therapeutic effects of *P. histicola* supplementation on inflammatory arthritis [18], multiple sclerosis [19], and type 1 diabetes [20]. So far, no data on the sleep-promoting effects of this probiotic supplementation have been reported. According to our unpublished data from human microbiome studies, we have observed that sleep duration and quality are associated with the proportion of *Prevotella*.

In this study, we aimed to investigate the effects and dose–response relationship of human gut commensal *P. histicola* supplementation on sleep architecture in rats compared with those of *P. stercorea* supplementation. We evaluated the effects of different *P. histicola* strains on sleep and pooled the results to understand the general effects of this species rather than limiting our findings to a particular strain. Furthermore, we evaluated the effects of *P. histicola* extracellular vesicle (EV) administration on sleep architecture.

## 2. Materials and Methods

### 2.1. Bacterial Culture and Preparation of the Probiotic Suspension of the Prevotella Species

In this study, we used *P. histicola* and *P. stercorea*, which are anaerobic and gram-negative bacteria. In particular, we investigated four *P. histicola* strains, including T05-04, CD3:32, Korean Collection for Oral Microbiology (KCOM) 4227 (accession no.: KCCM13105P), and KCOM 3796 (accession no. KCCM13103P), designated in the present study as “Strain 1”, “Strain 2”, “Strain 3”, and “Strain 4”, respectively. Strain 1 was isolated from human oral squamous cell carcinoma tissue and Strain 2 from human small intestinal tissues (biopsy samples for celiac disease). Strains 3 and 4, which were recently deposited in Korean Culture Center of Microorganisms (KCCM; Seoul, Republic of Korea), were isolated from human saliva by the KCOM (Gwangju Metropolitan City, Republic of Korea). In addition, we used one *P. stercorea* strain isolated from human feces, CB35 (JCM 13469), as a reference because, together with *P. copri*, this is the most relatively abundant *Prevotella* species in the human gut.

The organisms were inoculated onto anaerobic blood agar plates and cultured at 37 °C for 24 h in an anaerobic environment. The organisms were identified by 16S rDNA gene sequence analysis. Two forms of probiotics, live bacterial suspensions in liquid media for *P. stercorea* and *P. histicola* Strain 1 and Strain 2 and live freeze-dried bacteria for *P. histicola* Strains 3 and 4, were prepared. To prepare the freeze-dried probiotics, the culture supernatant was harvested after centrifugation (10,000× *g* for 10 min at 4 °C), re-suspended with sterile phosphate-buffered saline (PBS), and freeze-dried.

For the daily administration to the experimental animals, we adjusted the liquid medium probiotics to a final concentration of 10^10^ colony-forming units (CFU)/mL, while the freeze-dried probiotics were re-suspended with sterile distilled water to a final concentration of 10^10^ CFU/mL for Strains 3 and 4 and 10^6^ CFU/mL for Strain 4. Probiotics were stored in a BD GasPakTM EZ Pouch (BD, Franklin Lakes, NJ, USA) at 4 °C until administration. All bacterial culture and probiotic preparation procedures were conducted by the KCCM and KCOM, which were equipped with an anaerobic culture system. The KCCM confirmed bacterial gene sequences which were assessed by a commercial laboratory.

### 2.2. Preparation of the P. histicola EV Suspension

The EV suspensions were prepared with the bacterial cultures of Strains 1 and 3 in a commercial laboratory (Institute of MD Healthcare Inc., Seoul, Republic of Korea) following a previously reported method [21]. The harvested culture supernatant, which included EVs, was passed through a 0.22 µm filter to eliminate intact bacterial cells. The filtered supernatant was concentrated using a hollow cartridge (100 kDa molecular weight cutoff). The concentrated supernatant was passed again through a 0.22 µm vacuum filter to remove remaining cells. Afterwards, EVs were pelleted by ultracentrifugation (150,000× *g* for 3 h at 4 °C). To obtain purified EVs, pellet EVs were resuspended in PBS, layered in a discontinuous sucrose gradient solution, and centrifuged (at 200,000× *g* for 20 h at 4 °C). We resuspended the final pellets in PBS and stored them at −80 °C prior to use.

### 2.3. Animals and Experimental Conditions

A total of 46 7-week-old Sprague–Dawley male rats weighing 220–250 g (DBL Co., Ltd., Eumseong, Republic of Korea) were used in this study. All rats were individually housed in conventional cages with a 12-h light/12-h dark photoperiod (lights on 09:00–21:00) at a constant temperature (24 ± 2 °C). The rats were allowed ad libitum access to standard chow (RodFeed; DBL Co.) and water in their home cages. We treated all animals used in this study in accordance with the National Institutes of Health’s Guide for the Care and Use of Laboratory Animals, and the entire procedure was approved by the Animal Ethics Committee of Jungwon University (Approval no.: JWU-IACUC-2022-2).

Figure 1 shows the rats assigned to different experimental groups. We assigned the animals to each of the following nine groups: a no administration group (*n* = 5) that did not receive bacteria during the entire experiment, *P. stercorea* group for rats (*n* = 5) receiving live *P. stercorea* during the administration period, and *P. histicola* groups for rats (*n* = 26) receiving live *P. histicola* during the administration period.

The Strain 1 and Strain 2 groups included five rats each; the Strain 3 group had six rats. We pooled the sleep recording data of the three strains for analysis. The Strain 4 group was divided into two additional subgroups, Dose 1 and Dose 2 groups, each comprising five rats, according to the quantity of bacteria administered. For the *Prevotella* groups, except for the Dose 1 group, each rat received 10^10^ CFU/day live bacteria during the 7-day administration period. The Dose 1 group received 10^6^ CFU/rat/day over the same period.

We assigned 10 rats to the EV group receiving either Strain 1 or Strain 3 EV suspensions; each rat received EV extracted from 10^8^–10^9^ CFU daily. We pooled the sleep recording data of EV group for analysis regardless of strain type. We used the complete sleep recording data from 42 rats for analysis. Data from four rats were not obtained because of the disconnection of the recording devices (Figure 1A).

We administered 1 mL distilled water to all rats during the adaptation period. Further, the rats received one vehicle, 1 mL of sterile distilled water (for *P. histicola* Strains 3 and 4) or bacterial culture medium (for *P. stercorea* and *P. histicola* Strains 1 and 2) or PBS (for *P. histicola* EVs), during the baseline period. Both *P. histicola* and *P. stercorea* groups received live bacteria suspended in a vehicle during the 7-day administration period and subsequently a vehicle only during the bacteria withdrawal period (Figure 1B). For the EV groups, we obtained the sleep recording data during the 1-day baseline and 6-day administration periods without the withdrawal period. We administered the vehicles, probiotic suspensions, and EV suspensions using an oral gavage.

### 2.4. Surgery and Monitoring Method for Sleep Architecture

We implanted the animals with electroencephalography (EEG) and electromyography (EMG) electrodes affixed to a head-mount (Pinnacle Technology Inc., Lawrence, KS, USA) for continuous recording of the sleep–wake profiles. Under isoflurane anesthesia (induction 5%, maintenance 2–2.5%), two pairs of EEG electrodes were screwed into the frontoparietal and lateral lobes of the skull. A pair of EMG electrodes was imbedded in each animal’s dorsal neck muscle. These electrodes were connected to a preamplifier for EEG and EMG recordings via a flexible cable system, which allowed the animals to move freely. We recorded the EEG and EMG signals daily during the baseline, administration, and withdrawal periods and amplified these on a digital polygraph (Sirenia Sleep Pro; Pinnacle Technology Inc.) at 200 Hz. We set the filter ranges for EEG and EMG at 0.3–30 and 10–200 Hz, respectively.

After collecting data for the 42 rats, we analyze the 24-h sleep–wake profiles over 1–2 days of the baseline period, 3 days (Days 1, 3, and 6 for the EV groups; Days 1, 4, and 7 for the other groups) of the administration period, and 3 days of the withdrawal period (Figure 1B). We analyzed the 10 s epochs in the EEG and EMG recordings using sleep analysis software (Sirenia Sleep Pro, version 1.6.1), which allowed for manual scoring of the remaining unscored epochs after automated scoring on the basis of visual inspection.

We defined sleep latency as the time from the beginning of recording to the appearance of the first three consecutive 10 s non-rapid eye movement (NREM) sleep epochs. For each animal, we calculated the cumulative times per day of wake, rapid eye movement (REM) and NREM sleep, and sleep latency.

### 2.5. Statistical Analysis

Descriptive statistics were used to calculate mean and standard deviations (SD). Nonparametric tests, such as the Wilcoxon signed rank test and Kruskal–Wallis test, were used because of the small sample size of each group. We compared the differences in sleep–wake profiles among the groups. Changes in sleep–wake profiles during the administration and withdrawal periods were evaluated for comparison with the baseline data.

To evaluate a dose–response relationship, we pooled the 3-day recording data during the bacteria administration period and compared sleep time among the groups. For nonparametric multiple comparisons, we used the Dwass–Steel–Critchlow–Fligner method. All testing was based on a 2-sided level of significance (*p* < 0.05) and conducted using SAS software (SAS 9.4., SAS Institute, Cary, NC, USA).

## 3. Results

### 3.1. Effects of Live Prevotella Bacteria Administration on Sleep Architecture

Table 1 shows the means and SD for total sleep time, REM and NREM sleep time, and wake time from the 24 h sleep–wake profiles in each group. We observed no significant differences in baseline data, except for NREM sleep data, among the groups. REM sleep time recorded on Day 1 of bacteria administration was significantly different among the groups (*p* < 0.05); especially, the *P. stercorea* group showed shorter REM sleep time than the no-administration group. We observed no significant differences among the groups on Days 4 and 7 of bacteria administration and Days 1 and 3 of bacteria withdrawal. However, on Day 2 of bacteria withdrawal, total sleep (*p* < 0.01), NREM sleep (*p* < 0.05), and wake time (*p* < 0.05) were significantly different among the groups; the *P. histicola* group showed longer total sleep and NREM sleep time and shorter wake time than those of other groups. When compared with the baseline sleep durations, the *P. histicola* group showed significantly increased total sleep (*p* < 0.001), REM sleep (*p* < 0.05), and NREM sleep times (*p* < 0.01) and decreased wake time (*p* < 0.001) on Day 7 of bacteria administration, and we observed similar findings on Day 4 of bacteria administration and during the period of bacteria withdrawal. In strain-specific analyses, we observed increased sleep time with borderline significance, regardless of strain type, during *P. histicola* administration (data available on request). In the other groups, no significant differences occurred from the baseline.

Figure 2 depicts the changes in time from the baseline sleep–wake profile in a given time point for each group. Figure 2A shows total sleep time changes; we observed no significant changes in the no-administration and *P. stercorea* groups. For the *P. histicola* group, total sleep time gradually and significantly increased after bacteria administration (*p* < 0.05); there was a 51.5 min increase on Day 7 of administration from the baseline and an increasing trend sustained until Day 3 of bacteria withdrawal (mean time increase: 63.2 min on Day 2). Figure 2B shows the changes in wake time. We observed significant decreases until Day 3 of bacteria withdrawal in the *P. histicola* group (*p* < 0.01).

Figure 2C,D present REM and NREM sleep time changes, respectively. The *P. histicola* group showed significantly increased NREM sleep time on Day 7 of bacteria administration (mean time increase: 38.9 min) and during the bacterial withdrawal period (mean time increase: 48.3 min on Day 2) compared with those of the baseline (*p* < 0.01). Moreover, REM sleep time also increased significantly on Days 4 and 7 of bacteria administration (*p* < 0.05) and on Day 2 of bacteria withdrawal as compared with those of the baseline (*p* < 0.01).

Figure 3 shows a 24 h cumulative latency time of the three groups. We observed no significant changes in the no administration and *P. stercorea* groups. In the *P. histicola* group, latency time decreased significantly on Day 1 of bacteria administration (mean time decrease: 17.4 min) and on Days 2 and 3 of bacteria withdrawal compared with that of the baseline (*p* < 0.05).

### 3.2. Dose–Response Effects of Live P. histicola Administration on Sleep Architecture

Based on the significant results in Figure 2 and Figure 3, we further evaluated the dose–response relationship for the *P. histicola* administration. For comparison with the no-administration group, we pooled the 3-day sleep–wake profile data analyzed during the bacteria administration period and calculated its differences with the baseline data. We observed an increasing trend in total sleep time for the three groups, such as none, 10^6^ CFU, and 10^10^ CFU. In particular, the Dose2 group showed a significant increase (mean time increase: 48.7 min) compared with the no-administration group (*p* < 0.05, Figure 4A). Similarly, we observed an increasing trend in NREM sleep time across the groups; the Dose2 group had a significantly increased NREM sleep time (mean time increase: 40.1 min) compared with that of the no-administration group (*p* < 0.05, Figure 4C), but we found no significant differences in REM sleep time (Figure 4B).

### 3.3. Effects of P. histicola EV Administration on Sleep Architecture

Figure 5 shows the effects of EV administration on sleep status. We observed an increasing trend in total sleep and NREM sleep time, with borderline significance on Day 3 of EV administration compared with that of the baseline (*p* = 0.05) in the NREM sleep data.

## 4. Discussion

In this study, we analyzed sleep architecture data and evaluated the effects of probiotic supplementation with *P. histicola* and *P. stercorea* on rat sleep–wake profiles. Compared with the baseline sleep duration, *P. histicola* administration significantly increased sleep duration (52 min for total sleep, 13 min for REM sleep, 39 min for NREM sleep) on Day 7 of administration. The improved sleep state remained after supplementation withdrawal. Probiotic-derived *P. histicola* EV also increased NREM sleep time. However, the *P. stercorea* group showed no significant effects on sleep. An increasing trend was observed in the dose–response relationship between the *P. histicola* group and total and NREM sleep duration.

Data on the effects of probiotics on the sleep state are limited. In two clinical trials, healthy adults with the supplementation of probiotic mixtures including *Lactobacillus* and *Bifidobacterium* species showed decreased Pittsburgh Sleep Quality Index (PSQI) scores, indicating improved sleep quality, compared with that of a placebo group [22,23]. However, the species responsible for the beneficial effects on sleep quality remained unclear because of the use of probiotic mixtures.

Two studies have shown the sleep-promoting effects of a single *Lactobacillus* strain [12,13]. Mice subjected to a pentobarbital-induced sleep test on Day 14 showed significantly increased sleep time and decreased sleep latency after the administration of a specific *L. fermentum* strain for 14 days [12]. Another study analyzing sleep–wake profiles using the same probiotics found longer NREM sleep time in the supplementation groups compared with that of a control group under an insomnia-induced environment [13]. Furthermore, two studies evaluated the supplemental effects of a single strain of *L. gasseri* or *B. longum* on the PSQI scores of students under potential stress due to academic exams and found improved sleep quality compared with those of a placebo group [14,15]. All these studies used a specific strain for sleep-promoting effects.

In our study, we used one strain derived from the small intestine and three strains from the oral cavity to assess their sleep-promoting effects. Not only the pooled data but also the individual data of *P. histicola* strains showed an increase in total sleep time during the administration period. T05-04, designated as Strain 1, which is a *P. histicola* strain originating from the human oral cavity [24], is known to have similar DNA sequences of a novel strain isolated from the human intestine [17].

Bacterial EV molecules are nanometer-sized spherical particles measuring < 300 nm in diameter and contain proteins, lipids, and genetic materials [25]. Our results showing that *P. histicola* EV administration increased NREM sleep time suggest that *P. histicola* itself, rather than other gut bacteria that *P. histicola* influences, has a sleep-promoting potential.

GABA production can be a potential biological mechanism underlying the sleep-promoting effects of *P. histicola* supplementation. Some *Lactobacillus* and *Bifidobacterium* species have been suggested as GABA-producing bacteria because they contribute to GABA production by increasing glutamate decarboxylase (GAD) activity under acidic conditions resulting from the end-products of fermentation [26]. Because *P. histicola* can utilize carbohydrate producing acidic products [24], it may be involved in the biosynthesis of GABA via the GAD pathway. Most serotonins in the human body are synthesized in the gut by commensal microbiota, particularly spore-forming bacteria [27]. Although *P. histicola* is a non-spore-forming bacterium, it may be implicated in the regulation of the circadian rhythm involving serotonin conversion and melatonin synthesis [28]. It may be worthwhile to evaluate the role of *P. histicola* in the kynurenine pathway (KP) and the serotonin pathway based on the evidence of the relation of KP and *P. histicola* with multiple sclerosis [18,29].

Both the strengths and limitations of our study must be considered in the interpretation of our findings. Although we used several strains of *P. histicola* as probiotic supplements, the sample size of each group was small, and the supplementation duration was short. A lack of the gut microbiota composition data for the experimental animals might be another limitation. In our intervention study not yet reported, however, we conducted gut microbiota composition analyses and observed increased proportions of *Prevotella* species in participants who consumed prebiotic supplements. In an earlier report of ours, these participants showed improved sleep quality due to prebiotic supplementation [30]. Based on the results of our human studies, we investigated the direct effects of *Prevotella* species on sleep in the current animal experiment. The evaluation of biological mechanisms related to sleep-promoting effects was limited in the present study.

In summary, this animal study showed that probiotic supplementation with human-gut-derived *P. histicola* can increase total sleep and NREM sleep time and decrease wake time and sleep latency. Further research is warranted to confirm the sleep-promoting effects of *P. histicola* supplementation, evaluate its safety, and explore the biological mechanisms underlying the effects.

## Figures and Tables

**Figure 1 microorganisms-11-01151-f001:**
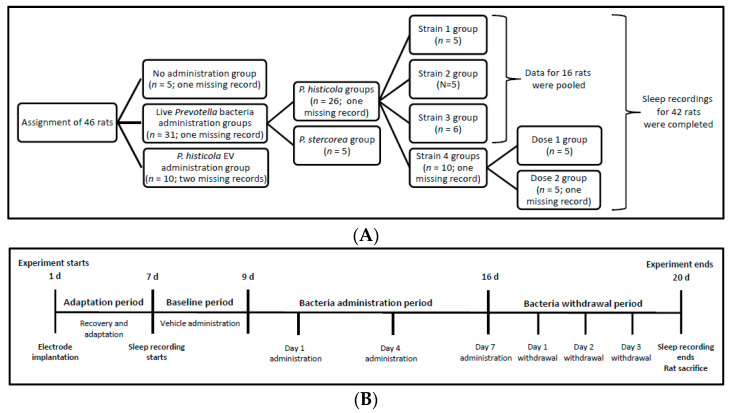
Schematic of animal group assignment and experimental design. (**A**) animal group assignment; (**B**) experimental design. EV, extracellular vesicle.

**Figure 2 microorganisms-11-01151-f002:**
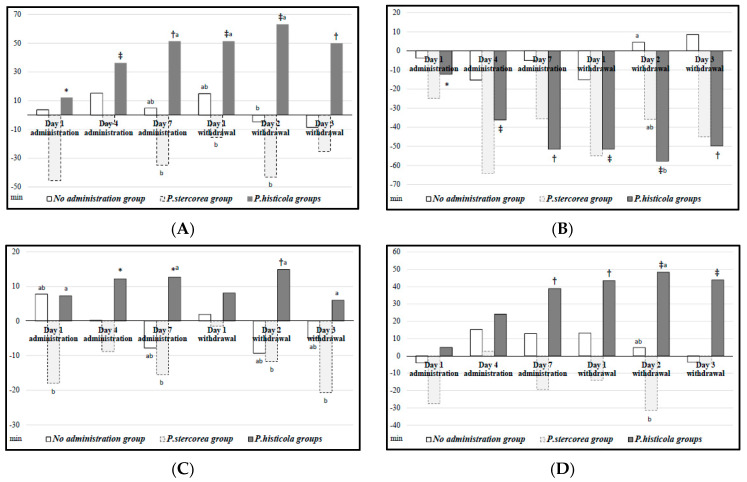
Sleep–wake profiles of rats administered with human-gut-derived live *Prevotella* bacteria. The y-axis indicates changes in time from the baseline sleep–wake profile in a given time point for the no-administration group (*n* = 4), the *P. stercorea* group (*n* = 5), and the *P. histicola* group (*n* = 16). *p* values (symbols *, †, and ‡ indicate <0.05, <0.01, and <0.001, respectively) were derived from the Wilcoxon signed rank test in comparison to the baseline. Different letters indicate statistical significance (*p* < 0.05) for multiple comparisons between groups. REM, rapid eye movement; NREM, non-rapid eye movement. (**A**) Total sleep time; (**B**) Wake time; (**C**) REM sleep time; (**D**) NREM sleep time.

**Figure 3 microorganisms-11-01151-f003:**
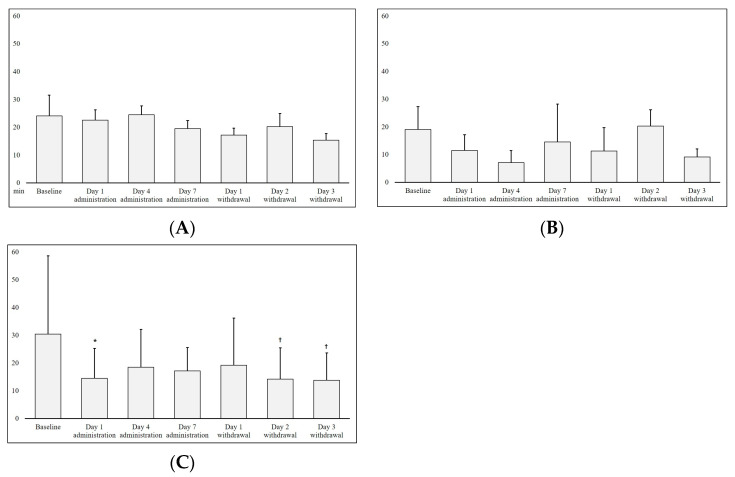
Cumulative latency time from the sleep–wake profile of rats administered with the human-gut-derived live *Prevotella* bacteria. *p* values (symbols * and † indicate <0.05 and <0.01, respectively) were derived from the Wilcoxon signed rank test in comparison to the baseline. (**A**) No administration group; (**B**) *P. stercorea* group; (**C**) *P. histicola* group.

**Figure 4 microorganisms-11-01151-f004:**
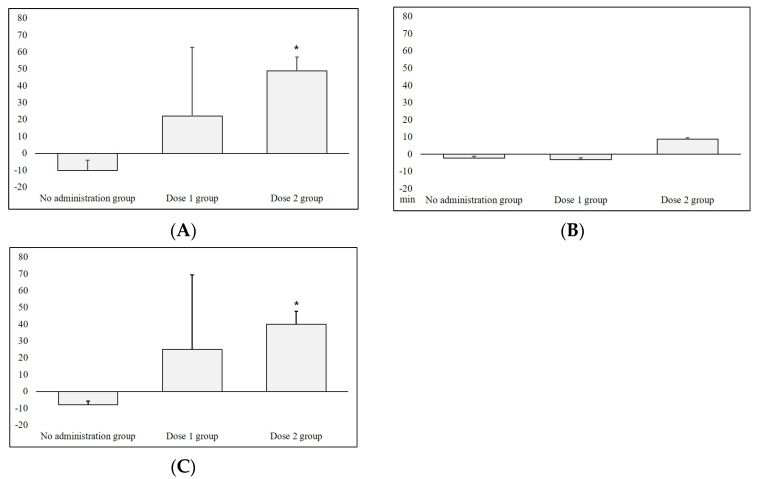
Dose-response effects in the sleep-wake profile in rats receiving human-gut-derived live *P. histicola* Strain 4. After pooling the 3-day sleep-wake profile data collected during the bacteria administration period, we calculated changes in time from the baseline sleep-wake profile, depicting these in the graph: no administration group (*n* = 4), Dose 1 group (*n* = 5), Dose 2 group (*n* = 4). We derived *p* values (* indicates < 0.05 for pairwise comparisons) from the Kruskal−Wallis test. REM, rapid eye movement; NREM, non-rapid eye movement. (**A**) Total sleep time; (**B**) REM sleep time; (**C**) NREM sleep time.

**Figure 5 microorganisms-11-01151-f005:**
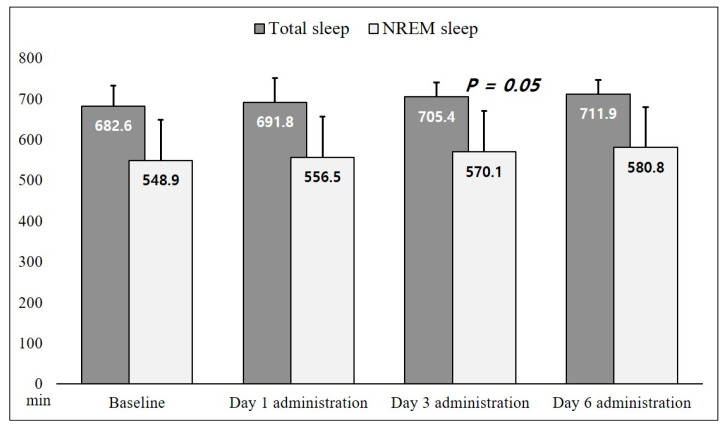
Total sleep and NREM sleep time from the sleep–wake profile in eight rats receiving *P. histicola* extracellular vesicles. We derived the *p* value from the Wilcoxon signed rank test in comparison with the baseline. NREM, non-rapid eye movement.

**Table 1 microorganisms-11-01151-t001:** Comparison of 24-h sleep–wake cycle profiles between rats not receiving bacteria (no administration group) and two animal groups (*P. stercorea* and *P. histicola* groups) receiving human-gut-derived live *Prevotella* bacteria.

Sleep-Wake Profiles	No Administration Group	*P. stercorea* Group	*P. histicola* Group	*p* Value ^(1)^
(Number of rats)	(*n* = 4)	(*n* = 5)	(*n* = 16)	
Vehicle administration baseline ^(2)^				
Total sleep time, min	708.8 ± 23.3	729.4 ± 42.6	678.1 ± 38.8	0.067
REM sleep time, min	133.9 ± 6.3	127.8 ± 26.5	117.5 ± 14.4	0.079
NREM sleep time, min	574.8 ± 19.1 ab	601.6 ± 20.6 a	560.6 ± 34.9 b	0.044
Wake time, min	671.6 ± 23.4	721.2 ± 123.7	702.1 ± 38.8	0.353
Bacteria administration, Day 1				
Total sleep time, min	712.5 ± 20.1	683.8 ± 33.3	690.4 ± 50.1	0.580
REM sleep time, min	141.7 ± 10.7 a	109.8 ± 12.0 b	124.8 ± 16.6 ab	0.028
NREM sleep time, min	570.8 ± 21.6	574.1 ± 36.0	565.6 ± 40.0	0.774
Wake time, min	667.8 ± 20.2	696.3 ± 33.2	689.8 ± 50.0	0.580
Bacteria administration, Day 4				
Total sleep time, min	724.1 ± 36.9	723.3 ± 41.1	714.4 ± 54.6 *	0.846
REM sleep time, min	134.1 ± 10.2	119.0 ± 20.3	129.7 ± 21.4 *	0.453
NREM sleep time, min	590.0 ± 28.3	604.3 ± 53.4	584.7 ± 37.7	0.758
Wake time, min	656.3 ± 36.8	657.0 ± 41.1	665.9 ± 54.6 *	0.846
Bacteria administration, Day 7				
Total sleep time, min	713.7 ± 25.1	694.6 ± 28.6	729.6 ± 39.2 ‡	0.167
REM sleep time, min	126.1 ± 3.2	112.3 ± 23.7	130.2 ± 20.7 *	0.435
NREM sleep time, min	587.7 ± 23.3	582.3 ± 20.3	599.5 ± 27.4 †	0.231
Wake time, min	666.6 ± 25.1	685.6 ± 28.6	650.6 ± 39.2 ‡	0.167
Bacteria withdrawal, Day 1				
Total sleep time, min	723.7 ± 27.5	713.9 ± 22.4	729.6 ± 39.2 †	0.746
REM sleep time, min	135.8 ± 4.6	126.3 ± 27.5	125.6 ± 22.6	0.704
NREM sleep time, min	588.0 ± 27.3	587.6 ± 13.9	604.1 ± 41.6 †	0.329
Wake time, min	656.5 ± 27.4	666.3 ± 22.4	650.6 ± 49.5 †	0.746
Bacteria withdrawal, Day 2				
Total sleep time, min	704.2 ± 33.6 ab	686.2 ± 19.2 b	741.3 ± 38.8 ‡a	0.007
REM sleep time, min	124.6 ± 8.1	116.1 ± 18.9	132.4 ± 30.2 †	0.358
NREM sleep time, min	579.6 ± 30.3 ab	570.1 ± 16.0 b	608.9 ± 24.8 ‡a	0.010
Wake time, min	676.1 ± 33.5	685.4 ± 12.3	644.3 ± 33.7 ‡	0.017
Bacteria withdrawal, Day 3				
Total sleep time, min	700.3 ± 42.7	704.0 ± 48.2	728.0 ± 50.8 †	0.458
REM sleep time, min	129.0 ± 10.3	107.1 ± 28.2	123.5 ± 18.8	0.328
NREM sleep time, min	571.2 ± 38.1	596.9 ± 30.8	604.5 ± 39.3 ‡	0.210
Wake time, min	680.1 ± 42.7	676.2 ± 48.0	652.3 ± 50.9 †	0.458

^(1)^ *p* values were derived from the Kruskal–Wallis test to compare values among the three groups. Different letters indicate statistical significance (*p* < 0.05) for multiple comparisons between groups. ^(2)^ Vehicle (saline or culture medium) was administered for 2 days and the average of 2-day recordings was calculated. *p* values (symbols such as *, †, and ‡ indicate <0.05, <0.01, and <0.001, respectively) were derived from the Wilcoxon signed rank test for comparison with the baseline value. REM, rapid eye movement; NREM, non-rapid eye movement.

## Data Availability

The data presented in this study are available on request from the corresponding author.

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
