# Peer review of "Oral Administration of Human-Gut-Derived Prevotella histicola Improves Sleep Architecture in Rats"

_microorganisms, 2023, doi:10.3390/microorganisms11051151_

Round 1
Reviewer 1 Report
The paper by Daewui Yoon and Inkyung Baik investigated the Oral administration of human gut-derived Prevotella effect on rat sleep architecture.
Statistics need to be run with multiple correction tests and some part need to be better explained.
Lines 48-49 The paper introduction deserves a connection between dysbiosis and inflammation status, which would presumably be present in all of the cited pathologies.
How much different in terms of genomes similarity are P. stercorea and P. histicola investigated strains? This part should be deepened in the introduction in order to pave the ground for the reason of the author's choice and support
Apart from belonging to the Bacteroidetes order it is not clearly stated why Prevotella should be the eligible genus to be investigated.
Table 1: I would suggest clarifying the meaning of group in the table header.
For all the statistical analyses I would suggest correcting the comparison significances by multiple tests.
Figure 1: also here I would suggest correcting for multiple tests.
Author Response
Ref. No.: microorganisms-2307915
Title: Oral administration of human gut-derived Prevotella histicola improves sleep architecture in rats
We are most grateful with the reviewers’ comments on the above-referenced manuscript. We have listed here our response to each comment.
Reviewer 1’s comments:
1. Statistics need to be run with multiple correction tests and some part need to be better explained. For all the statistical analyses I would suggest correcting the comparison significances by multiple tests. Figure 1: also here I would suggest correcting for multiple tests.
Response: Thank you for the reviewer’s comments. We have now described nonparametric multiple comparison method (lines 332-333, revised) and added significance (letters) for multiple comparisons in the results (Tables 1 and Figure 1, revised).
2. Lines 48-49 The paper introduction deserves a connection between dysbiosis and inflammation status, which would presumably be present in all of the cited pathologies.
Response: We have now added more information (lines 46-49, revised)
3. How much different in terms of genomes similarity are P. stercorea and P. histicola investigated strains? This part should be deepened in the introduction in order to pave the ground for the reason of the author's choice and support. Apart from belonging to the Bacteroidetes order it is not clearly stated why Prevotella should be the eligible genus to be investigated.
Response: We agree on the reviewer’s comments regarding the insufficient rationales of the use of Prevotella species in this study. We have now added explanation why Prevotella was selected (lines 52-54, revised). We confirmed P. stercorea and P. histicola by 16S rDNA gene sequence analysis, but did not compare the genome similarity of these two species. On the basis of earlier data of phylogenetic tree for Prevotella species [Balakrishnan et al. Frontiers In Immunology. 2021:12;609644][Alauzet et al. International Journal of Systematic and Evolutionary Microbiology. 2007;57:2216.], P. stercorea and P. histicola show a genetic distance.
4. Table 1: I would suggest clarifying the meaning of group in the table header.
Response: We have now revised it (Table 1).

Reviewer 2 Report
The study investigated the sleep-promoting effects of gut microbiota in rats receiving P. histicola, P. stercorea, or extracellular vesicles (EV). The P. histicola group showed increased total sleep, REM sleep, and NREM sleep time during administration and withdrawal periods. Linear trends in the dose-response relationship for total and NREM sleep were observed in the P. histicola group. Administration of EV increased NREM sleep time on the 3rd day. The study suggests that P. histicola may have potential as a sleep aid. Although it is interesting, there are several major concerns that has to be addressed before consideration publication.
Introduction.
It would be better to introduce the significant of P. histicola and P. stercorea, and it potential implications in humans or animals.
Methods
The EVs isolation steps are not clear, and not easy to follow up.
Only male mice were included in this study, why? And if all these mice are from littermate?
Results
How do you determine the amount of EVs injection? And what is difference between bacteria injection and EVs injection?
Author Response
Ref. No.: microorganisms-2307915
Title: Oral administration of human gut-derived Prevotella histicola improves sleep architecture in rats
We are most grateful with the reviewers’ comments on the above-referenced manuscript. We have listed here our response to each comment.
Reviewer 2’s comments:
1. Introduction: It would be better to introduce the significant of P. histicola and P. stercorea, and it potential implications in humans or animals.
Response: We agree on the reviewer’s comments regarding the insufficient rationales of the use of Prevotella species in this study. We have now added explanation why Prevotella was selected (lines 52-54, revised). Because data regarding P. histicola and P. stercorea associated with disease are limited, we have now added information for other Prevotella species (lines 46-49, revised).
2. Methods: The EVs isolation steps are not clear, and not easy to follow up.
Response: We have now revised the EVs isolation method (lines 252-259, revised).
3. Only male mice were included in this study, why? And if all these mice are from littermate?
Response: We used male Sprague-Dawley (SD) rats only in this study to avoid the physiological variability and potential hormonal effects associated with the estrous cycle of female rodents. In this study, all SD rats were outbred in a commercial company; random mating was done to maintain genetic variability and regular genetic monitoring was conducted by Taconic Inc. confirming genetic variability.
4. Results: How do you determine the amount of EVs injection?
Response: To the best of our knowledge, no animal studies have investigated the sleep-promoting effects of EVs from any bacterial species, including P. histicola. We determined the amount of EVs considering the amount of live P. histicola (1010 CFU/mL) showing significant sleep-promoting effects in our study. Considering the loss of activity during resuspension of the lyophilized P. histicola, the final dose was determined as the amount of EVs extracted from 108~109 CFU/mL, which corresponds to the protein concentration of 2 mg~4 mg/mL approximately.
5. What is difference between bacteria injection and EVs injection?
Response: We wanted to clarify whether a substance inside P. histicola has sleep-promoting effects or whether a substance synthesized or induced by bacteria has. On the basis of our results regarding EVs, the sleep-promoting effects might be due to a substance inside P. histicola. Further studies with more extensive analytical experiments are warranted to identify this EV constituent.

Round 2
Reviewer 1 Report
This reviewer thinks that all the required modifications have now been taken into account. The paper can be accepted for publication in its present form.
Author Response
Ref. No.: microorganisms-2307915
Title: Oral administration of human gut-derived Prevotella histicola improves sleep architecture in rats
Reviewer 1’s comments: This reviewer thinks that all the required modifications have now been taken into account. The paper can be accepted for publication in its present form.
Response: Thank you for the reviewer’s suggestion regarding the acceptance of our manuscript for publication. The manuscript has been checked by a professional English editor.